# XTM: A Novel Transformer and LSTM-Based Model for Detection and Localization of Formally Verified FDI Attack in Smart Grid

Anik Baul [1,*], Gobinda Chandra Sarker [2], Pintu Kumar Sadhu [1], Venkata P. Yanambaka [3] and Ahmed Abdelgawad [1]

1   College of Science and Engineering, Central Michigan University, Mount Pleasant, MI 48858, USA
2   Department of Electrical and Electronic Engineering, Mymensingh Engineering College, Mymensingh 2200, Bangladesh
3   College of Sciences, Texas Woman's University, Denton, TX 76204, USA
*   Correspondence: baul1a@cmich.edu

**Abstract:** The modern smart grid (SG) is mainly a cyber-physical system (CPS), combining the traditional power system infrastructure with information technologies. SG is frequently threatened by cyber attacks such as False Data Injection (FDI), which manipulates the states of power systems by adding malicious data. To maintain a reliable and secure operation of the smart grid, it is crucial to detect FDI attacks in the system along with their exact location. The conventional Bad Data Detection (BDD) algorithm cannot detect such stealthy attacks. So, motivated by the most recent deep learning (DL) developments and data-driven solutions, a new transformer-based model named XTM is proposed to detect and identify the exact locations of data intrusions in real-time scenarios. XTM, which combines the transformer and long short-term memory (LSTM), is the first hybrid DL model that explores the performance of transformers in this particular research field. First, a new threshold selection scheme is introduced to detect the presence of FDI, replacing the need for conventional BDD. Then, the exact intrusion point of the attack is located using a multilabel classification approach. A formally verified constraints satisfaction-based attack vector model was used to manipulate the data set. In this work, considering the temporal nature of power system, both hourly and minutely sensor data are used to train and evaluate the proposed model in the IEEE-14 bus system, achieving a detection accuracy of almost 100%. The row accuracy (RACC) metric was also evaluated for the location detection module, with values of 92.99% and 99.99% for the hourly and minutely datasets, respectively. Moreover, the proposed technique was compared with other deep learning models as well, showing that the proposed model outperforms the state-of-the-art methods mentioned in the literature.

**Keywords:** smart grid (SG); false data injection (FDI); formal model; transformer; Long short-term memory (LSTM); attack detection; attack localization

## 1. Introduction

The Internet of Things (IoT) has become an indispensable part of our daily lives, integrating sensors, actuators, software, and other electronic devices directly into digital network systems [1]. The internet revolution is occurring to a great extent with the help of IoT, which finds application in almost every sphere of our daily lives, from home surveillance to power station monitoring [2]. The electrical power system is one of the most critical elements of a country's economic and social development. The research community is relentless in finding innovative technologies for future grid systems because of the high consumption and rising demand of electricity at the commercial, residential and industrial levels. There are many vulnerabilities in the traditional grid system, which means it fails to fulfill the growing demand while lacking new generation features such as self-healing, real-time pricing, congestion management, reliability and security [3]. Additionally, the

penetration of a large amount of renewable energy into the grid and the emergence of electric vehicles are making the existing old-fashioned grid systems vulnerable. In contrast, the next-generation power grid, also known as the Smart Grid (SG), has been improved significantly from the traditional grid system, offering all the new features mentioned earlier. SG provides a more secure, fast, reliable, and efficient power system operation by integrating modern information and communication technologies (ICTs) with the existing conventional grid [4]. The rapid advancement of SG is mainly facilitated by the combination of upgraded control techniques and advanced information systems [5]. IoT allows SG to transfer a significant amount of data between users and operators, making it more vulnerable to cyber attacks. STUXNET, which targeted Iran's nuclear plant in Natanz, was one of the first known cyber attacks to cause physical damage [6]. In 2015, cyber attacks on Ukraine's power grids caused prolonged, widespread power outages [7]. It is seen from the reports of US ICS-CERT and Kaspersky ICS-CERT that the energy sector is considered the most vulnerable among the other network infrastructures [8]. These cyber attacks can threaten the operational reliability of power systems by damaging vital system parameters.

Cyber attacks in SG can be broadly categorized into three different types, namely physical, communication and information attacks [9]. Considering the other attacks false data injection (FDI) is one of the most common and lethal attacks [10]. The goal of FDI is to manipulate the state variables by introducing malicious data into the initial measurement set [11]. Figure 1 depicts how an FDI attack can happen in SG. In a real power system, SG has three major parts: power generation, transmission, and distribution [12]. Remote terminal units (RTU) are used to transfer data from physical devices to the master control room, from where all the decisions have been taken. Supervisory Control and Data Acquisition (SCADA) is a crucial part of the control center that continuously monitors and regulates all the parameters of SG systems by collecting real-time data to run the operation of SG efficiently [13]. For simplicity, we have shown only one central control system. It can be seen from the figure that an attacker can launch a successful FDI attack at any part of the SG system.

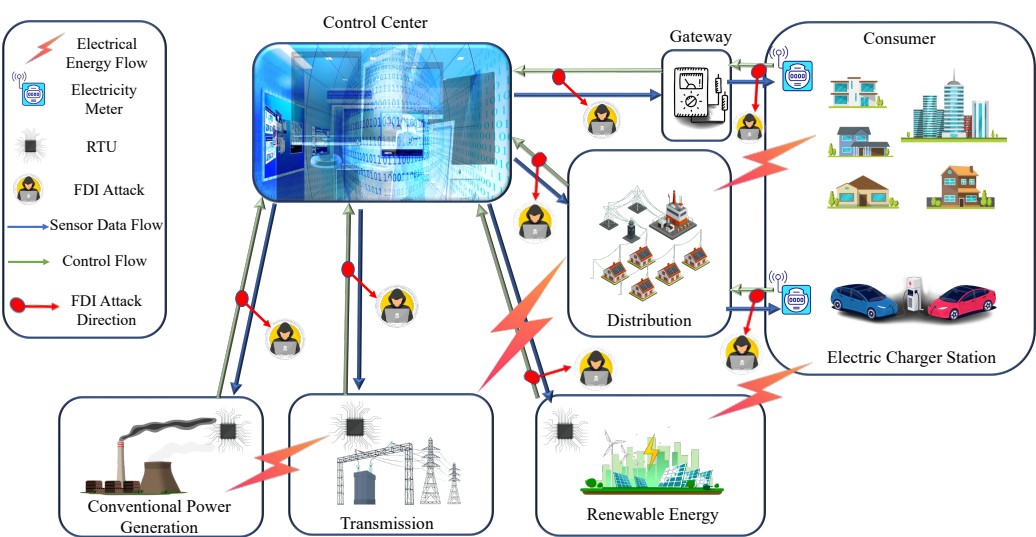

**Figure 1.** FDI Attack in Smart Grid.

A well-structured FDIA can circumvent the need for the conventional BDD method and manipulate the state estimates of a power system. The intrusion of a false state value in a system branch can lead to uncontrollable excessive or inadequate energy production, which may result in catastrophic system collapse if it goes undetected. The best way to combat FDI is to continuously monitor a system to identify and isolate any unusual activity happening in the system. Substantial research has been conducted on FDIA detection, with various strategies

suggested, which can be broadly divided into two categories: model-based and data-driven detection algorithms [14]. While data-driven detection algorithms are independent of system parameters, model-based methods need system parameters, which is the most significant limitation of this approach. Data-driven methodologies learn directly from training data, as opposed to deriving an algorithm from a predetermined attack and model of that system.

Numerous studies have been carried out to detect the presence of FDI in SG. In addition to FDI presence detection, the location of the attack is a crucial parameter to deploy appropriate countermeasures quickly. Therefore, researchers have focused on accurately identifying the location of FDI intrusion and isolating the compromised sensors using various machine learning algorithms in recent years [15]. Taking inspiration from natural language processing (NLP), we investigated the performance of a transformer algorithm to detect FDI in this paper, which has not been done previously in the literature. Based on transformer and LSTM, a novel algorithm, which we have named XTM, is proposed in this study for the accurate detection of FDIA along with the entry points of malicious data into the system using real-time monitoring of the power grid's physical signals. The model is first trained on the benign historical dataset to learn the internal structure of the data and to forecast the sensor measurements. To effectively detect the presence of anomalous behaviors, the anomaly detection technique needs precise predictions on the real-time data [16]. To find FDI in the system, a threshold level is considered, and the output of the real-time data prediction is fed as the input of the localization module that works as a multilabel classifier. In this paper, various deep learning models along with XTM are described to evaluate the proposed model. Earlier studies in the literature primarily focus on attack detection and localization. However, details on the attack vector generation process and the viability of the dataset utilized to train the model are most often insufficient. To bridge this problem, explicit instruction for building an attack vector using a formal model is provided in our study. From an existing work, we obtained our hourly data and based upon this we have made our minute-based dataset. As the formal model is time-independent, the attack vectors can be used for both the cases, such as hourly and minute basis data. To summarise, the main contributions of this work are as follows:

- We have introduced a novel data-driven Transformer and LSTM-based model named XTM to detect and localize FDI attacks in the smart grid. XTM is the first model that utilizes the Transformer algorithm in this particular research field.
- The proposed XTM algorithm is able to forecast sensor measurements in real time due to its independent nature of system parameters. So, it can be used as a tool to mitigate FDI attack in case of an intrusion in the power grid.
- Comparative analysis of four distinct deep learning models along with our proposed XTM is studied to determine the best multi-label classifier to detect the exact location of FDI attack. All of the tests were conducted using the IEEE 14 bus test system, where our proposed model outperforms others.
- We have discussed the data set and attack vector in detail, which is missing in most of the research works. We have used three different types of data sets: one benign hourly basis, one benign minute basis and one is attack vector data set. The hourly dataset used in this project was adopted from Shahriar et al.'s work [17] and the attack vector was taken from [18]. Then based on this, we have made our dataset which is minute based. The proposed model is intensively tested on both of these data sets, where it shows better performance than the other methodologies. The attack vector, hourly and minutely datasets are made publicly available in the Github repository (https://github.com/gcsarker/XTM, accessed on 16 December 2022).

The rest of the article is organized as follows: Section 2 provides a brief literature review that motivated our work. In Section 3, preliminary information is discussed in detail. The proposed model, along with the frameworks' technical specifications are discussed in Section 4. Evaluation procedure is described in Section 5, while in Section 6, experiments and results are shown along with the brief description of the dataset. In Section 7, different case studies have been conducted. Finally, the paper is concluded in Section 8.

## 2. Related Works

Recently advancements in both machine learning (ML) and deep learning (DL) models have motivated many researchers to work on the detection and localization of FDI attacks in smart grid. In this section, some recent studies regarding FDI in SG are reviewed that led to our work. In [19], a generative-adversarial based semi-supervised learning framework (GBSS) is used to identify attacks under challenging circumstances, where the samples are partially labeled. The proposed semi-supervised algorithm incorporated with CGAN is compared with 18 other different scenarios of two identical transmission systems, and this model outperforms the other techniques. Lei et al. [12] proposed a gated recurrent neural network (GRNN) model based on the classification of predicted residuals (CPRs) for the instantaneous forecasting of measurement data from where they got the residuals. Then, they used a wavelet-transform-based convolutional neural network (CNN) to improve detection accuracy by separating the anticipated residuals from the incorrect data. An actual microgrid testbed is used to verify the proposed model. A computationally efficient extra trees classifier model is suggested at [20] to accurately identify FDI attacks even in low sparsity. A stacked auto-encoder consisting of two auto-encoders is used to solve the high dimensionality problem. Pei et al. [21] detected FDI by using a clustered partitioning state estimation (CPSE) technique. They mainly used a deviation-based detection method (DBDM) based on an additional Kalman filter. However, to improve the detection accuracy this method would need to consider the variation in load and generation over time. Zhang et al. [22] employed a semisupervised approach, combining auto-encoders and GAN to identify FDI in smart grid. For evaluation purposes, they compared their proposed models with other semi-supervised learning techniques where the detection accuracy of the suggested approach is high and reliable. However, this detection technique only applies to the same power system topology. In another study [16], a forecasting-aided anomaly detection framework using a CNN-LSTM-based auto-encoder is proposed to detect FDI in the meter data management system. A threshold level is considered, when observing the consumption trends to determine if any malicious data are mixed in. They only used three months of consumption data but lack of providing details information about the data set and the attack vector. It also does not account for this factor if the system becomes more extensive with a bulk feature vector.

Roy et al. [23] depicted how different DL models perform, when identifying FDI attacks in the automatic generation control systems (AGCS). Five distinct DL-based models have been evaluated to identify the malicious data. No details about the attack have been discussed here. When unidentified threats or novel attacks are offered during predictions, the performance of the models decreases significantly. The authors in [24], proposed extreme learning machine (ELM) optimized with artificial bee colony (ABC) algorithm. They used an auto-encoder to reduce the dimension of measurements. Although the proposed mechanism outperforms classifiers like SVM, RBF and naive bayes, it can only detect the presence of FDI. Camana et al. [25] proposed extra-trees algorithm, where the high dimension space of measurements is reduced by kernel principal component analysis (KPCA) technique. The proposed scheme was able to achieve accuracy of 98.35% accuracy in detecting the presence of FDI, compared to other feature selectors and ML models. Identifying the presence of FDI was the main goal of the previous studies. However in recent years, researchers have shifted their focus to both detecting the presence and the exact location of FDI. Nagaraj et al. [26] considered the temporal nature of power system states and applied Ensemble CorrDet with Adaptive Statistics (ECD-AS) that is shown to outperform ML models like k-nearest neighbor (KNN), support vector classifier (SVC) and gaussian naive bayes (GNB). However, the simulation of FDI is not properly discussed. The proposed mechanism also requires a separate classifier for each bus. Although the time dependency of states is considered, the dataset is generated from one day only. Mukherjee et al. [27] did a comparative investigation of the performances of different AI techniques along with the traditional BDD to detect and localize FDI in smart grids. Here, CNN, CNN-LSTM, CNN-GRU and KNN are used for real-time, precise location detection of data

intrusions, and among all the other models, CNN performs the best. Rashed et al. [28] used the weighted least square (WLS) method along with minimum variance unscented Kalman filter (MVUKF)to detect FDI. They also located the malicious data intrusion point by partitioning the whole power system into small sections. However, in this work, they only consider one cyber-attack at a time. In [29], LSTM with a Temporal Convolutional Neural Network (TCN) based model is proposed to identify the location of FDI as multi-label classes. It was mainly developed without using any statistical suppositions from the attack model. Five different datasets, along with individual L2-norms have been used in this work. Here they compare their proposed model with other-state-of the-art benchmark techniques such as CNN, LSTM, etc. where it outperforms the other models based on the locational detection accuracy. Wang et al. [30] proposed a convolutional neural network (CNN) in a multi-label classification fashion along with a standard bad data detector (BDD) to detect the position of FDI. The performance of the proposed model was evaluated under several noise and attack conditions, where it performed very satisfactorily. In this study [31], a model based on the fusion of a bi-directional gated loop unit and convolution neural network is suggested to address the problem of detection and identification of FDI. This model performs better than the traditional convolution neural network-based model. Table 1 summarizes previous studies' main contributions, models, limitations, etc., which inspired us to investigate more in this research field.

The states of the power system are temporal in nature, regardless many prior research developed algorithms without considering this. So, training model on historical sensor data can effectively increase model performance. As the size of the power system grows, so will the number of measurements. It is also notable that nowadays, handling time series data has become very efficient due to several neural network-based models such as LSTM, 1-D CNN, RNN, auto-encoders, etc. [23]. However, these models might fail to perform well on large number of measurements. On the other hand, the transformer is a DL model, that can learn large sequences due to multi-headed self-attention and simultaneous processing of all sequence components. This motivated us to explore and develop a transformer-based model which is fast, reliable and efficient in practical application for detecting FDI in SG. Moreover, very few works have revealed their data which is a crucial parameter. We have made our dataset publicly available with a comprehensive description.

**Table 1.** Recent Research on FDI attack in smart grid.

| Authors | Objectives | Model Used | Is Dataset Available? | Limitations |
|---|---|---|---|---|
| Lei et al. [12] | FDI presence detection | GRU with wavelet transform-CNN | Yes | • Attack location cannot be determined<br>• Only attack magnitude is considered, did not discuss attack sparsity |
| Mahi et al. [16] | FDI presence detection | Auto-encoder + CNN-LSTM-based auto-encoder | No | • Attack location is not considered<br>• Unable to handle power system with large number of features<br>• Did not account the seasonal variation of measurements |
| Nagaraj et al. [26] | FDI presence detection | ensemble CorrDet with adaptive statistics (ECD-AS) | No | • Requires separate detector for each buses<br>• Measurements taken from only one day |
| Camana et al. [25] | FDI presence detection | KPCA with Extra-Trees algorithm | No | • Attack sparsity is not considered<br>• Cannot detect which measurement sensor is compromised |
| Roy et al. [23] | FDI presence detection | LSTM-based three models + Auto-encoder based two models | No | • Attack sparsity is not considered<br>• Unable to handle the novel attacks<br>• Detailed information about the data set and the attack vector lacking |
| Mukherjee et al. [27] | FDI location detection | CNN | No | • Did not account the seasonal variation of measurements<br>• Attack case study targeting particular measurements did not consider the dependency with other measurements |

## 3. Preliminaries

The relevant background knowledge that helps in explaining the XTM framework is included in this section.

### 3.1. Power System Model

The AC power flow analysis is the most widespread and significant analysis for a power system. It depicts the relationship between complex bus voltages and currents flowing via the interconnected lines attached to those buses [32]. However, a complete AC power flow simulation is computationally expensive. Moreover, it raises the problem of non-linearity, which is very complex. On the other hand, DC power flow analysis has been used more frequently for the real-time dispatch of power systems due to its simplicity and reliability. In this work, the DC power flow model is used to analyze the attacks against state estimation where only the voltage phase angles are handled as variables, and the voltage magnitudes at all buses are retained constant at 1 per unit (PU). In our work, We consider a DC linearized state estimation method. A mathematical technique known as state estimation (SE) analyzes the raw measurement data obtained from the remote sensors to estimate system states. In the power system, SE is used to get the voltage and phase angles of all the existing buses. SE is built on the power flow equations, state variables, and the measurements vector [29]. In the SCADA system, SE is one of the crucial parameters. Optimal Power Flow is fed by the output of the state estimator. This state estimator calculates magnitudes of the voltages with phase angles, transmission line flows, and bus loads. The commonly used weighted least square-based SE is susceptible to stealthy attacks, in which a malicious party might change specific measurements to corrupt the estimator's solution. The following equation shows the relationship between the measurement vector z and the states vector x:

$$z = h(x) + e \tag{1}$$

Here *h(x)* is the measurement function that shows the connections between the measurement set and the states vectors and e is the measurement error vector. As we have considered linear DC power-flow assumptions and the equation becomes,

$$z = Hx + e \tag{2}$$

where *H* represents the Jacobian matrix. The number of potential measurements m is significantly greater than the number of states n. The following equation is used to perform linear estimation

$$\hat{x} = (H^T W H)^{-1} H^T W z \tag{3}$$

$\hat{x}$ is the system states vector when measurement errors have a zero mean distribution, and W represents the diagonal matrix. Therefore, after the estimation $\hat{z} = h(\hat{x})$, here $\hat{z}$ is the estimation of measurement. Residuals between measurements and estimated states $|z - \hat{z}|$, are used to detect malicious data. Here $\tau$ is defined as the threshold level of residual. It is considered that there is malicious data if $|z - \hat{z}| \geq \tau$.

### 3.2. FDI Attack

Many factors, including meter malfunctions and malicious attacks, may introduce inaccurate measurements. Bad data are typically caused by random system failures in the measurement equipment. The traditional BDD cannot detect FDI attacks. An adversary can contaminate the estimator's solution by adding erroneous data to the measurement set. An attacker intentionally creates and injects fake data into the system while aiming for an FDI attack. It is important to note that an FDI attack can only adjust loads of two or more buses; it cannot increase the system's overall load. [33]. In an FDI attack, the primary purpose of the attacker is to change the state variable of the system by adding some malicious data. The BDD will fail if the measurement set z contains deliberately inserted false data a where

$a = Hc$. In this case, c represents the added value that has been created after the injection of amount, a to the state estimate, x. Here the attack maintains stealthiness as the injected data will not be visible in the residual that we can understand from the following equations,

$$residual = \parallel (z + a) - H(\hat{x} + c) \parallel = \parallel z - H\hat{x} \parallel \tag{4}$$

However, if the attackers are aware of the measurement matrix H, they can deliberately mix the malicious data so that the residual of the original measurement vector remains the same as the residual of the measurement vector along with the attack vector. However, a successful stealthy FDI attack can be happened without having full knowledge of topology matrix, H. In [34], a realistic FDI with partial or inadequate knowledge is presented where the attacker does not have the real time information of different power system equipment. Moreover, [35] demonstrates that even without being aware of the topology, an attacker can still launch an FDI attack.

### 3.3. CNN

Convolutional deep neural networks in contrast to traditional neural networks, learn the localized pattern in an input sequence. The input to the 1-d CNN model has $n$ input sequence representing $n$ timesteps, each having multivariate features. A set of 1-d kernels, also known as filters of fixed window size are chosen. The convolution operation between the kernels and the input sequence produces output features. This is a process where kernels of window size $k$ slide over the sequence with fixed strides. These feature maps encode the response of a filter pattern at different locations of the input sequence. If we consider $L$ convolutional layer, the first convolution operation near the multivariate input sequence $z$ is shown in Equation (5) [30].

$$C_{1,j} = ReLU(Z^* h_{1,j} + b_{1,j}) \tag{5}$$

where $C_{1,j}$ is the output feature map of the first convolutional layer. The convolution operation is denoted by (*). The $J^{th}$ kernel is represented as $h_{1,j}$. $b_{1,j}$ is the bias term added to the output. Rectified linear unit (ReLU) is adopted to encode non-linearity. So, the $J^{th}$ feature space of the $I$th convolutional layer $C_I$, it can be written as.

$$C_{I,j} = ReLU(C_{I-1,j}^* h_{I,j} + b_{I,j}) \tag{6}$$

We need the previous feature maps and kernels to produce the feature map of the $I$th convolutional layer. The pooling operation is performed for computational efficiency and to capture the totality of the input sequence by down-sampling the dimensionality of the feature maps. Maxpooling involves sliding small windows over the feature maps and finding the maximum value.

### 3.4. LSTM

Recurrent neural network (RNN) is a type of neural network architecture that specializes in sequence learning tasks. Long short-term memory network (LSTM) is a special type of RNN first introduced in [36]. Simple RNN loops through the sequence components and only considers the previous sequence component to process the current timestep, and hence facing the gradient vanishing issue explained in [37]. LSTM overcomes the limitation by introducing a memory cell state which may carry information across many timesteps, thus always maintaining a state of past information. Figure 2 shows the structure of LSTM for a single timestep t. The cell states ($C_{t-1}, C_t$) run parallel to the hidden states ($h_{t-1}, h_t$) in which the input sequence is fed at each timestep. In LSTM, the interaction with cell state is achieved by three mechanisms to determine which sequence component to keep, forget or update [38]. These mechanisms are expressed as follows.

$$f_t = \sigma(W_f[h_{t-1}, x_t] + b_f) \tag{7}$$

$$i_t = \sigma(W_i[h_{t-1}, x_t] + b_i) \tag{8}$$

$$\tilde{C} = \tanh(W_c[h_{t-1}, x_t] + b_c) \tag{9}$$

$$C_t = f_t * C_{t-1} + i_t * \tilde{C} \tag{10}$$

$$o_t = \sigma(W_o[h_{t-1}, x_t] + b_o) \tag{11}$$

$$h_t = o_t * \tanh(C_t) \tag{12}$$

where $x_t$, $h_{t-1}$ represents the input sequence component of the current timestep and the hidden state from the previous timestep respectively. The forget, input, and output gates are denoted by $f_t$, $i_t$, and $O_t$ respectively. The value of each of them is found through the activation of the sigmoid function that outputs between 0 and 1. The forget gate $f_t$ determines how much cell state information needs to be removed. $i_t$ along with $\tilde{C}$ calculates what portion of the input at the current timestep needs to be stored and the amount of update required in the cell state. The new cell state is denoted by $C_t$. The weight matrices $W_f$, $W_i$, $W_c$ and $W_o$ are learned during the training of LSTM to optimize gate behavior. $b_f$, $b_i$ and $b_o$ are scaler values added as bias.

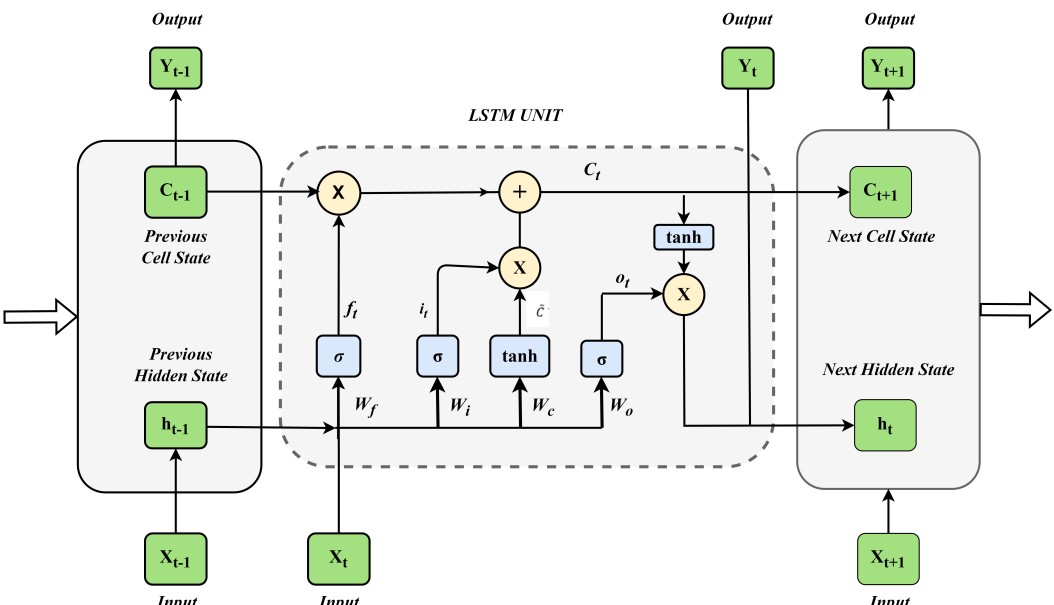

**Figure 2.** Block diagram of lstm for a single time step.

### 3.5. Transformer

The transformer [39] is a DL model introduced in 2007 by Vaswani et al. Although the original paper demonstrated the efficacy of machine translation, it is extensively applied in many sequence learning tasks, even outside the scope of natural language processing [40,41]. This is brought about by the transformer model overcoming the limitation of the previously existing models as well as outperforming them in many areas [42]. Though somewhat mitigating compared to the conventional recurrent neural network, LSTM still suffers from the vanishing or exploding gradient if the sequence to be processed is long. LSTM acts following Markov property, where the information in a timestep depends only on hidden states of past timesteps. So, parallel processing of input sequences cannot be realized in LSTMs. The advantage of a transformer is that it processes the whole input sequence at once, replacing recursion with multi-headed self-attention. The attention mechanism in general consists of three components, namely the queries, keys, and values. In an encoder-decoder architecture, the previous decoder output is denoted by $q$, where the keys and values would be encoded input denoted by $k$ and $v$. In the case of dot product attention, a score value $(e_{q.k_i})$ is computed as a dot product of query and key. The scores are passed

through a softmax function to generate weights, denoted as $\alpha_{q.k_i}$. Finally, the weighted sum of the value vector is calculated as shown in Equations (13)–(15) [43].

$$e_{q.k_i} = q.k_i \tag{13}$$

$$\alpha_{q.k_i} = softmax(e_{q.k_i}) \tag{14}$$

$$attention(q, k, v) = \sum_i \alpha_{q.k_i}.v_k \tag{15}$$

The transformer encoder takes advantage of self-attention, where queries ($Q$), keys ($K$) and values ($V$) are created from the encoder input. Also, the dot product of query and key are scaled by the root of the dimension of key vectors denoted as $d_k$ for computational stability. This is shown in Equation (16).

$$Attention(Q, K, V) = softmax\left(\frac{QK^T}{\sqrt{d_k}}\right) \tag{16}$$

$$Multihead(Q, K, V) = concat(head_1, head_2, .........., head_h)W^0 \tag{17}$$

$$head_i = Attention(QW_i^Q, KW_i^K, VW_i^V) \tag{18}$$

Rather than implementing a single attention function, the transformer uses a mechanism called multi-headed attention. Here multiple attention function is carried out in parallel. These attention heads are concatenated and multiplied with an output weight matrix $W^0$, as shown in Equations (17) and (18). multi-headed attention provides the additional benefit of allowing the attention layer to have multiple representation sub-spaces [39]. For our purpose, we employ the transformer encoder architecture as shown in the block diagram in Figure 3.

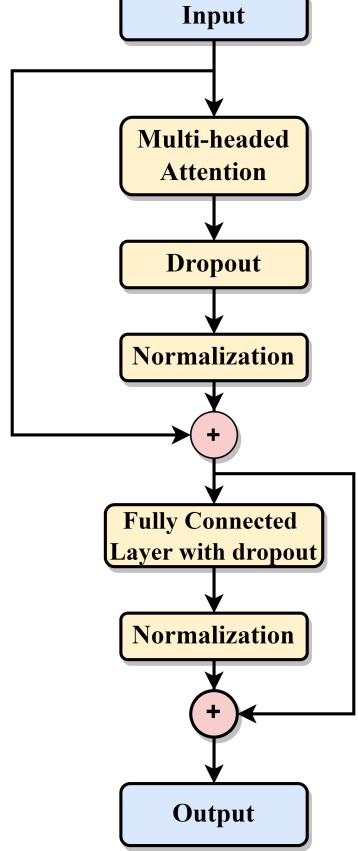

**Figure 3.** Block diagram of transformer encoder for the proposed model.

## 4. Technical Details

In this study, we have explored contemporary and advanced deep learning architectures for the purpose of detecting the presence and location of FDI attacks. This section elucidates the details of our proposed approach, which consists of three modules: the formal module for the attack vector, the FDI presence detection module, and the location detection module. The First module describes the attack vector and its different attributes. The second module detects the presence of FDI and estimates the state variables to mitigate the impact of the attack. The localization of the attack is executed in the third module. The block diagram representation of the proposed technique is illustrated in Figure 4.

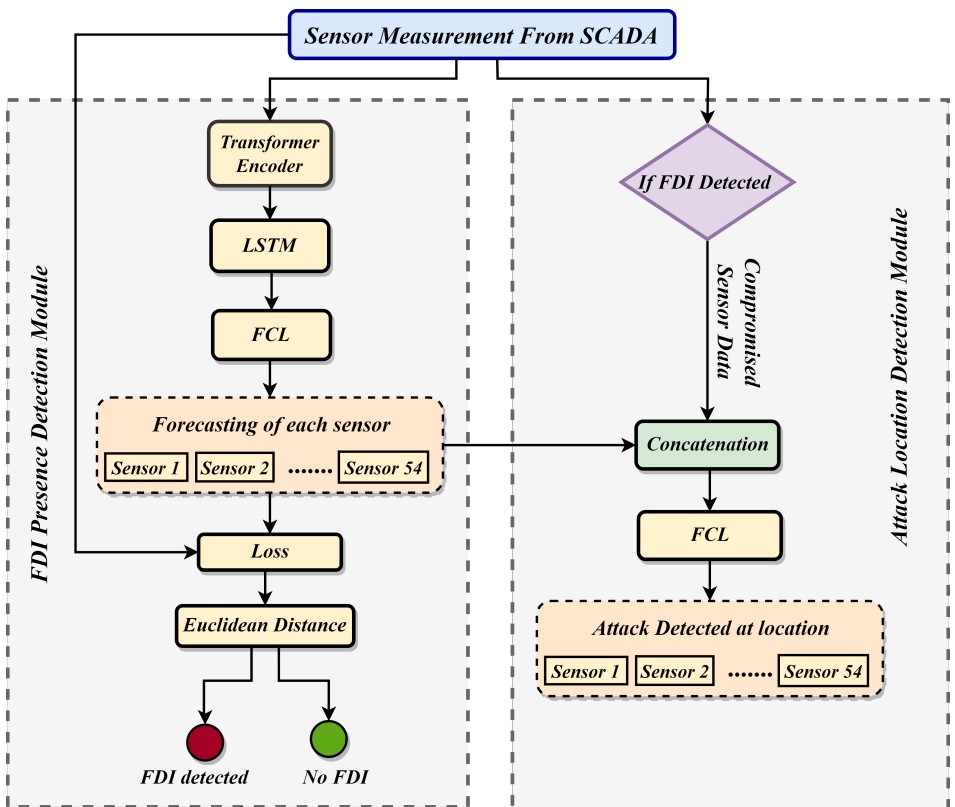

**Figure 4.** Block diagram of transformer encoder for the proposed model.

### 4.1. Formal Module

FDI attacks can be conducted on a variety of power grid components, including sophisticated metering infrastructure, transmission, and distribution systems [44]. The FDI attacks are referred to be non-stealthy if the injected data are sufficiently large so that the traditional error detection system can easily pick them up. In the non-stealthy instance, the attackers simply create random attack vectors and tamper with the sensor readings because they are unaware of the measurement matrix H. This paper explores FDI attacks on the static state estimate in the AC power transmission system. The stealthy attack vector used in our model is mainly taken from existing work [18]. A formal verification-based framework is used to construct the attack vector of our model. A constraint satisfaction problem (CSP) is built by using all the system variables, such as the voltage magnitudes along with their corresponding phase angles, transmission line flows, and the total bus system loads. RTU and intelligent electronic devices IEDs are used in the substations of different power systems. These devices collect data from the sensor and send it back to the central energy management system (EMS). Table 2 depicts various system characteristics and attacks attribute to describe FDI attacks on SE.

**Table 2.** Parameters for formal modeling.

| Notationl | Defination |
|---|---|
| b | Total number of buses in the topology |
| l | Total number of lines in the topology |
| $f_i$ | Starting bus of line $i$ |
| $e_i$ | Ending bus of line $i$ |
| $d_i$ | Admittance of line $i$ |
| $P_i^L$ | Total power flow in line $i$ |
| $P_j^B$ | Total power consumption in bus $j$ |
| $\theta_j$ | The measurement of state value for bus $j$ |
| $L_{j,in}$ | The incoming line sets towards bus j |
| $L_{j,out}$ | The outgoing line sets towards bus j |
| $P_j^G$ | Total power generated in bus $j$ |
| $P_j^D$ | Total Load power flow in bus $j$ |

In most cases, two sensors are used in a single transmission line to measure the forward and backward current flows, and only one sensor is used to measure the power consumption in a single bus. If any power system has l number of lines and b number of buses, then the total number of sensors equals m = 2l + b. An adversary can penetrate the system knowing the topology matrix. The primary purpose of an attacker is to manipulate the SE to form a stealthy attack which will lead to compromised system operation. Some formal constraints are considered while making this attack vector. After considering all these constraints, the attack vectors were constructed. The conditions are given below.

1. Is the measurement taken or recorded?
2. Is the measurement secured?
3. Does the attacker have the accessibility to manipulate the measurement?

Here, formal constraints to build a DC power system are described. Equation (19) demonstrates how transmission line power flow depends on the difference between line admittance, and bus phase angles. From (20) & (21), we can see that the net power at a bus, which is also called the bus's power consumption depends on the difference between the total power generation of a bus and the total load of that particular bus. In (20), we did not consider any other power losses as DC power only considers active power ignoring reactive power and transmission losses [45].

$$\forall_{1 \le i \le l} P_i^L = d_i(\theta_{f_i} - \theta_{e_i}) \tag{19}$$

$$\forall_{1 \le j \le b} P_j^B = \sum_{i \in L_{j,in}} P_i^L - \sum_{i \in L_{j,out}} P_i^L \tag{20}$$

$$\forall_{1 \le j \le b} P_j^B = P_j^D - P_j^G \tag{21}$$

If any attacker attacks bus j, that means the phase angle of that bus will be changed. To make a stealthy attack, an attacker can not deliberately compromise any random line power. An undetermined FDI attack has some crucial characteristics. Attackers will be able to make a stealthy FDI attack by maintaining the following conditions in the system.

1. Both the forward line and backward line power flows on each transmission line be equally compromised.
2. The net power of a bus should be zero.

The attack on state j indicates that there has been a change in the voltage phase angle at bus j. Equation (22) formalizes this condition. If $\Delta\theta_j$ is added in the phase angle of bus j, then the injected false data in the power flow measurement of line I can be calculated by using Equation (23). From Equation (25), we can calculate the amount of power consumption measurement, which needs to be changed because of the penetration of malicious data in the system. Attackers must change several measurements before launching an attack because they need to adjust various power line flows or consumption. In Equations (24) and (26),

line flows and power consumption is finalized by adding the specified malicious data. By analyzing an attack vector, it is possible to understand that the position and amount of malicious data need to be penetrated into the system. An attacker does not need to compromise all the sensors for a constructive FDI attack, instead focusing on the sensors having non-zero injection values.

$$\forall_{1 \leq j \leq n} c_j \rightarrow (\Delta\theta_j) \neq 0 \tag{22}$$

$$\forall_{1 \leq i \leq L} \Delta P_i^L = d_i(\Delta\theta_{f_i} - \Delta\theta_{e_i}) \tag{23}$$

$$\forall_{1 \leq i \leq L} \overline{P_i^L} = P_i^L + \Delta P_i^L \tag{24}$$

$$\forall_{1 \leq j \leq b} P_i^L = \sum_{i \in L_{j,in}} \Delta P_i^L - \sum_{i \in L_{j,out}} \Delta P_i^L \tag{25}$$

$$\forall_{1 \leq i \leq l} \overline{P_J^B} = P_j^B + \Delta P_j^B \tag{26}$$

*4.2. FDI Presence Detection Module (FPDM)*

Our proposed approach employs a hybrid of transformer-LSTM architecture that can detect the presence of such an attack very precisely. The input to this module is 48 h of past sensor measurements obtained through the supervisory control and Data Acquisition (SCADA) system. The input is fed to transformer encoder. The output of the transformer goes through two LSTM networks before finally obtaining the predicted output for the next hour through the fully connected layer prediction head. This method predicts all 54 sensor measurements for the next hour simultaneously. The error between the actual and predicted sensor measurements is calculated in the next step. To detect the presence of stealthy FDIA, the l2-norm of the error vector is used. The system is under attack if the l2-norm goes beyond a certain threshold $\tau$. The method of finding the threshold value is described in Section 5.2. This process essentially compares the estimated readings with the real readings of that hour. The output of this module is a flag indicating whether there is an attack or not. The additional benefit of this technique is that, the estimation of the state variables can be employed to mitigate the impact of FDIA. Various attack mitigation techniques have been proposed in the literature [46]. The authors of this article [47] have also adopted a similar approach to our research, using load forecast to drive automatic generation control (AGC) on power systems. This approach is proven effective because it can prevent the power system from going into hazardous situations allowing sufficient time for the operators to take control.

*4.3. Location Detection Module (LDM)*

To detect the attack location, we have adopted a multilabel classification technique. The goal is to simultaneously classify each sensor into two labels, whether it is under attack or not. Multilabel classification differs from usual classification in that it can assign multiple classes to a sample in contrast to classifying a sample into a single class from a set of fixed class labels. Our proposed method for detecting location involves looking at estimated values and the real values for each hour and figuring out which sensor is compromised. The idea is for the model to learn to detect which measurement is responsible for differing from the estimates. We have employed a multilayer perceptron network of 3 hidden layers for this task. The input to this module is the estimates from the FDI presence detection module as well as the sensor measurements, whereas the output is one for the compromised sensors, otherwise zero.

## 5. Evaluation Setup

This section describes the model training & testing setups along with other crucial parameters to describe our models performance.

### 5.1. Model Training

The proposed technique involves an attack presence detection module and a location detection module. To validate the performance of XTM in the presence detection module, we have also investigated four different models. The transformer encoder architecture of the proposed model is implemented, as illustrated in Figure 3. The transformer encoder block is followed by a stack of two long short-term memory layers of size 128. The sequential input is fed to the transformer encoder, and the output of the second LSTM layer is passed through a fully connected layer (FCL) of 128 nodes. The rectified linear unit function is applied to its output. The prediction of all state variables is taken from the multi-headed output with the number of heads equal to the maximum number of sensors. Each output head is a dense layer of only one node for estimating individual sensor values. The architecture of the Transformer-LSTM network is shown in Table 3. The pseudo-code for FPDM is shown in Algorithm 1. Here $S_t$ is the input time series of size T used during the training of FPDM. Each sample $X_t$ contains three types of measurements forward line flow, backward line flow and net power at 14 buses denoted by $f_t$, $r_t$ and $b_t$ respectively. The multivariate feature size M is the total number of measurements for each sample. The model is trained for a certain number of epochs and model parameters are updated after training the model on each minibatch $x(x\epsilon X)$ of size $B$. The binary classification output $F(F_0, F_1, \ldots, F_t)$ indicating the presence of FDI is determined after comparing the l2 norm of error between predicted output $y$ and input $x$ with the predefined threshold denoted as $\tau$. The location detection module receives two inputs, as illustrated in Algorithm 2. The first is the output prediction of the FPDM, which is denoted as $y$, and the second input is the sensor measurement data $S_t$ fed to the FPDM. These two inputs are concatenated and fed to a stack of 3 fully connected layers, each having 128 nodes. The ReLU activation function is applied to the output of each of these layers. The location of the attack is estimated from the final output $P(P_0, P_1, P_2, \ldots, P_t)$ of the location detection module in a multilabel manner. The final output is taken from a dense layer of node size equal to the number of input sensors, where each sensor is represented by the individual node. Sigmoid activation is applied to extract the probability of attack at each location.

**Table 3.** Proposed FDI presence detection model architecture.

| Stage | Type | Output Shape | Number of Params |
|---|---|---|---|
| 0 | Input | $48 \times 54$ | 0 |
| 1 | Multi-headed Attention | $48 \times 54$ | 11,880 |
| 2 | dropout | $48 \times 54$ | 0 |
| 3 | Normalize | $48 \times 54$ | 108 |
| 4 | Add with Input | $48 \times 54$ | 0 |
| 5 | FCL (ReLU) | $48 \times 128$ | 7040 |
| 6 | dropout | $48 \times 128$ | 0 |
| 7 | FCL (ReLU) | $48 \times 54$ | 6966 |
| 8 | Nomralize | $48 \times 54$ | 108 |
| 9 | Add with stage 4 output | $48 \times 54$ | 0 |
| 10 | LSTM (tanh) | $48 \times 128$ | 93,696 |
| 11 | LSTM (tanh) | 128 | 131,584 |
| 12 | FCL (ReLU) | 128 | 16,512 |
| 13 | Dropout | 128 | 0 |
| 14 | FCL (Linear) | $54 \times 1$ | 54 |

Total Number of Parameters: 274,860
Trainable Parameters: 274,860
Non-trainable Parameters: 0

---

**Algorithm 1:** FDIA presence detection algorithm

---

1 fontsize510

   **Input** $S_t$    : Multivariate time series sensor data $[X_0, X_1, \ldots, X_t]_T$

   **Sample** $X_t$ : $[f_0, f_1, \ldots, f_19, r_0, r_1, \ldots, r_19, b_0, b_1, \ldots, b_13]_M$

   **Output F**   : $[F_0, F_1, \ldots, F_t]_T$; $F_t = 1$ if FDI detected as 0

2 Initialize model weights, W;

3 **for** $i \leftarrow 1$ **to** *Num of Epochs* **do**

4     x ← generate training sequence of shape (B,T,M);

5     **for** $e \leftarrow 1$ **to** *Num of Encoders* **do**

6        mha ← MultiHeadAttention(x, Num of heads, head size);

7        k ← norm(mha) + x;

8        x ← RelU(FCL(k));

9        x ← norm(x) + k;

10     **end**

11     **for** $i \leftarrow 1$ **to** *Num of LSTM* **do**

12        calculate $f_t$ (Equation (7)),$i_t$ (Equation (8)),$\tilde{C}$ (Equation (9));

13        update cell state $C_t$ (Equation (10));

14        x ← calculate output $O_t$ and hidden states $h_t$;

15     **end**

16     x ← ReLU(FCL(x));

17     y ← Linear(FCL(x));

18     $W \leftarrow W - \alpha \frac{\delta L}{\delta W}$ updating model weights with SGD;

19 **end**

20 g ← l2norm|y-x|;

21 **if** $g \geq \tau$ **then**

22     F = 1

23 **end**

24 **else**

25     F = 0

26 **end**

27 Output F;

---

**Algorithm 2:** location detection module algorithm

---

1 fontsize510

   **Input** $S_t$    : Multivariate Sensor data $[X_0, X_1, \ldots, X_t]_T$

   **Output F**   : Multilabel output at time t, $[P_{0t}, P_{1t}, \ldots, P_{Mt}]_T$

2 Initialize $P \leftarrow []$;

3 Initialize Model2 weights $W_2$;

4 **for** $i \leftarrow 1$ **to** *Num of Epochs* **do**

5     x ← generate train sequence of shae(B,T,M);

6     y ← Model1(x);

7     x ← concatenate(y,x) **for** $i \leftarrow 1$ **to** *Num of FCL* **do**

8        x ← ReLU(FCL(x));

9     **end**

10     **for** $i \leftarrow 1$ **to** $M$ **do**

11        P ← append(sigmoid(FCL(x)));

12     **end**

13     Update weights : $W_2 \leftarrow W_2 - \alpha \frac{\delta L}{\delta W2}$

14 **end**

15 Output P;

### 5.2. Threshold Selection

One of the significant requirements of our proposed method is selecting a threshold to differentiate compromised measurements from benign ones. The threshold is chosen after the training of FPDM. The forecasting model is trained on normal sensor data to estimate the measurements at a particular time step. Our model is trained only on benign data sets. After the training procedure, the test set is copied and injected with attack vectors generated by the formal module. By applying these two sets of test data on our model, we obtained two sets of error vectors. One between the predicted response and real test set and the other with compromised test data.The errors are-

- error 1 = predicted data - real sensor data
- error 2 = predicted data - compromised sensor data

Then we calculate the vector magnitude of the error vectors using Equation (27). The distribution of error vector magnitudes is shown in the histogram Figure 5. The blue color bar graphs in Figure 5a depict the error distribution between the predicted data and compromised data for the strongest attack vector. On the other hand, Figure 5b describes the error distribution for the weakest attack vector, which is denoted by blue color bar graphs as well.

$$|E| = \sqrt{e_1^2 + e_2^2 + \ldots + e_M^2} \tag{27}$$

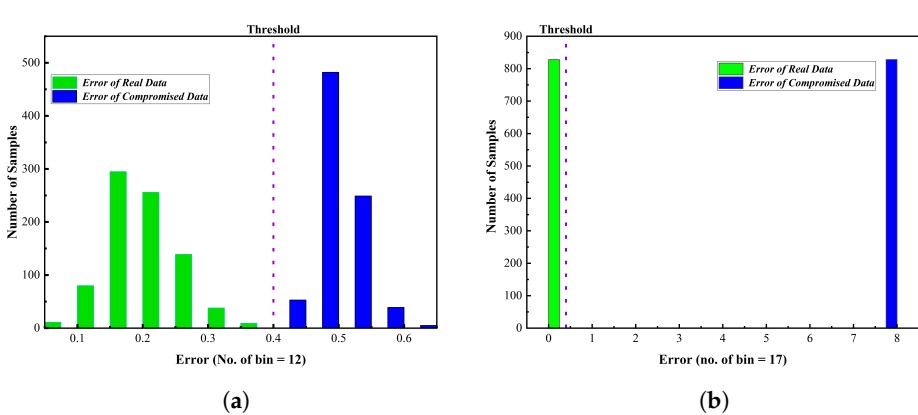

(**a**)    (**b**)

**Figure 5.** Error distribution : (**a**) for strongest attack vector (**b**) for weakest attack vector.

Table 4 shows the error magnitude of the benign test set at different percentiles. F1-score calculated using these percentiles as the threshold is also provided. The threshold is selected in a way that it can separate compromised and actual data with high accuracy. We recommend using the 99th percentile with some tolerance accounting uncertainty in the sensor data. A good threshold should be low so that it can detect even more stealthy attacks.

**Table 4.** f1-score at different threshold values.

| Percentile (%) | Error Magnitude | f1-Score |
|---|---|---|
| 95 | 0.3041 | 0.9732 |
| 96 | 0.31575 | 0.9783 |
| 97 | 0.32 | 0.9831 |
| 98 | 0.32365 | 0.98774 |
| 99 | 0.35368 | 0.9927 |
| 100 | 0.39788 | 0.99565 |
| Proposed $\tau$ | 0.4 | 0.9964 |

### 5.3. Minibatch and Cross-Validation

We have employed minibatch gradient descent for training both FPDM and LDM. This is adopted to enhance the convergence rate and avoid overfitting [48]. A batch size of 32 samples is selected for optimal performance. To properly evaluate model accuracy, we have split our training dataset into an 8:2 ratio. Half of the test set is taken as a validation set, whereas the other half is used to evaluate test performance. The batch size indicates the number of samples used by the model for gradient calculation at every parameter update. Each batch comprises a sequential sample of size 48. Since we predict the current time based on the previous 48 timesteps. Batches of data are prepared on the fly during the training procedure. Hence our model does not require much storage space and is very reliable to apply on a bigger dataset. In this research, We have experimented with five advanced deep learning models including the proposed method, namely CNN, CNN-LSTM, CNN-Transformer, Transformer and Transformer-LSTM (XTM). The superiority of CNN and CNN-LSTM in detecting false data injection has already been proven in the literature [27,29,30] . In our work, we have investigated how the performance increase by integrating the transformer with other algorithms. Every model is trained and evaluated on the Keras Tensorflow library using python programming language utilizing Keras Early-Stopping functionality. Among the models, CNN-LSTM, CNN-Transformer and XTM depict the highest accuracy. The evolution of training and validation loss over epochs for these models are shown in Figure 6a–c. The comparison of validation loss over the epoch among different models is illustrated in Figure 6d. It can be observed that the proposed algorithm showed the lowest and most stable validation loss. Figure 7 shows the training and validation loss of LDM, where the input estimates are taken from the XTM model.

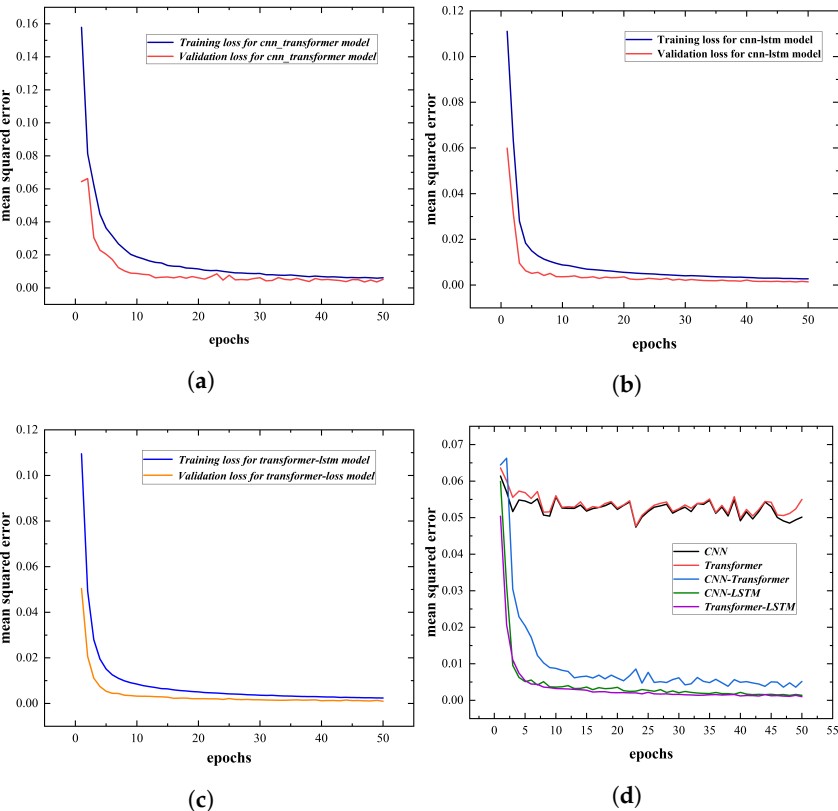

**Figure 6.** Loss vs Epoch Curve : (**a**) Training and validation loss curve of CNN-Transformer model (**b**) Training and validation loss curve of CNN-LSTM model (**c**) Training and validation loss curve of XTM model (**d**) Comparison of validation loss over epoch of different models.

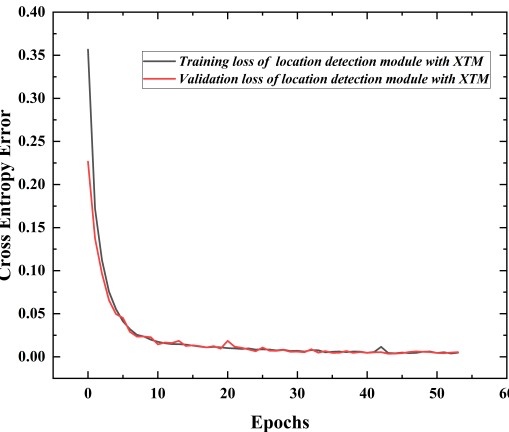

**Figure 7.** Loss Vs epoch curve during the training of LDM using estimates from XTM model.

### 5.4. Loss Function

Loss calculation between the ground truth and prediction value is of significant importance because it is used to update model parameters. Optimal parameters are achieved by minimizing the loss score. The model in the presence detection module minimizes mean squared error (MSE) loss between the predicted and actual sensor measurement as shown in Equation (28). However, the task in the location detection module is to estimate a probability vector indicating whether FDI is detected or not at every sensor position. So cross-entropy loss is applied between the prediction probability and target probability over a batch of size B as shown in Equation (29). Here, $M$ is the output size, in other words the number of sensors.

$$MSE(\hat{y}, y) = \frac{1}{B} \sum_{i=1}^{B} (\hat{y}_i - y_i)^2 \tag{28}$$

$$L(\hat{y}, y) = \sum_{b=1}^{B} -\frac{1}{M} \sum_{m=1}^{M} (y_i \log \hat{y}_i + (1 - y_i \log(1 - \hat{y}_i))) \tag{29}$$

## 6. Experiments and Results

The faithfulness of the proposed XTM algorithm is evaluated based on several experimental results. We have also conducted a comparative analysis with two other transformer-based models (Transformer, CNN-Transformer) and two commonly used deep learning models (CNN-LSTM and CNN), established for displaying superior performance in smart grid research. In this section, we discuss the findings of our study. Being an end-to-end approach, how the proposed algorithm can detect the presence of a false data injection attack, as well as demonstrate the attack location without relying on any other statistical methods such as BDD is presented.

### 6.1. Dataset

In this work, three different types of datasets such as hourly basis, minute basis and attack vector data are used. First of all, there are rarely any publicly available datasets that facilitate research on FDI in power system. The hourly dataset used in this paper was taken from [17]. They worked on FDI and gave open access to their dataset. As it is already a published work, we have decided to take our benign hourly time-series data from here which has 8760 samples along with 54 features. These 54 features represent different measurements in IEEE 14 bus system. Here the first 20 measurements depict the forward line power flow, the following 20 measurements represent the reverse power flow of these 20 lines, and the last 14 measurements indicate the 14 buses' power consumption. The attack vector dataset used in this project was adopted from [18]. This attack dataset was created by formal model on a standard IEEE-14 bus system which has 128 samples along

with 54 features. Our benign hourly data and attack vector datasets are made through formally verification which is proven to be mathematically accurate. Overall, these datasets perfectly aligned with this research, which is why we have chosen these already published data. Then We have decided to take the minute data because it is more granular than the hourly data and which is more practical in the real power system. The hourly dataset is extended utilizing linear interpolation, as shown in Equation (30). Interpolation estimates new values given a set of existing values. The hourly dataset is filled with interpolated measurement data at every one-minute interval. In the equation, $Y_1$ and $Y_2$ represent measurement data at any two adjacent hours $X_1$ and $X_1$. For any minute $x\epsilon(x_1, x_2, \ldots, x_{59})$ between $X_1$ and $X_2$ the sensor data are denoted by $y$. This data set has 525,600 samples and 54 features. The proposed model is intensively tested on both of these hourly and minute datasets where it shows better performance than the other methodologies.

$$y = Y_1 + (x - X_1)\frac{(Y_2 - Y_1)}{(X_2 - X_1)} \tag{30}$$

### 6.2. Performance Evaluation Metrics

To evaluate the attack detection accuracy of the FPDM, we have measured precision, recall and f1 score. These parameters are based on true positive (*TP*), true negative (*TN*), false positive (*FP*) and false negative (*FN*) values. In the context of attack presence detection, they can be formalized as follows.

- *TP*: Number of all correctly predicted compromised instances.
- *FP*: Number of incorrect predictions of uncompromised instances.
- *TN*: number of correctly predicted uncompromised instances.
- *FN*: number of incorrectly predicted compromised instances.

Precision measures the proportion of samples correctly classified among all the samples predicted as malicious as shown in Equation (31). The proportion of the samples classified as compromised out of all compromised samples determines the recall in Equation (32). F1-score is the harmonic mean of two scores as mentioned earlier delineating as a single score which is shown in Equation (33).

$$precision = \frac{TP}{TP + FP} \tag{31}$$

$$recall = \frac{TP}{TP + FN} \tag{32}$$

$$f1 = 2 \times \frac{recall \times precision}{recall + precision} \tag{33}$$

The forecasting accuracy of the FPDM is measured by calculating the loss between the actual and predicted state variables. Three matrices, namely root mean square error (RMSE), mean square error (MSE) and mean absolute error (MAE) is chosen for evaluation. The mathematical representation of RMSE, MSE and MAE is shown in Equation (28), (34) and (35). The number of samples is denoted by n.

$$RMSE = \sqrt{\frac{\sum_{i=1}^{n}(y_i - \hat{y})}{n}} \tag{34}$$

$$MAE = \frac{\sum_{i=1}^{n}(y_i - \hat{y})}{n} \tag{35}$$

### 6.3. FPDM Accuracy

The proposed XTM module is trained on uncompromised benign data to forecast state variables at each timestep. The forecasting accuracy of the XTM model is compared with other benchmark algorithms. RMSE, MSE, and MAE losses between the real and estimated state variables on the test set are shown in Table 5. Our proposed algorithm outperforms

the other models achieving the lowest scores in all the loss metrics. To definitively predict whether there is an attack, the l2 norm of the mse loss is calculated and compared with a threshold, as discussed in the threshold selection section. This prediction task is essentially a binary classification. Utilization of XTM allowed us to obtain the lowest threshold value of 0.4. The prediction accuracy using the XTM model has also proven superior over other techniques, as shown in Table 6. The proposed technique has achieved the highest macro average precision, recall, and f1-score on test data.

**Table 5.** Loss of forecasting module.

| Model | RMSE | MSE | MAE |
|---|---|---|---|
| CNN | 12.1323 | 276.8111 | 10.2596 |
| CNN-Transformer | 3.1835 | 16.7532 | 2.5116 |
| Transformer | 12.2238 | 281.6116 | 10.2524 |
| CNN-LSTM | 1.9447 | 6.0333 | 1.5055 |
| XTM (Transformer-LSTM) | 1.2635 | 2.4992 | 0.9866 |

**Table 6.** Performance comparison for detecting the presence of FDI in IEEE-14 bus system.

| Model | Threshold ($\tau$) | Precision | Recall | f1-Score |
|---|---|---|---|---|
| CNN | 1.25 | 0.7632 | 0.8129 | 0.7534 |
| CNN-Transformer | 1.00 | 0.9516 | 0.9547 | 0.9515 |
| Transformer | 1.25 | 0.7379 | 0.7989 | 0.7284 |
| CNN-LSTM | 0.4 | 0.9893 | 0.9893 | 0.9893 |
| XTM (Transformer-LSTM) | 0.4 | 0.9962 | 0.9962 | 0.9962 |

*6.4. Location Detection Module Accuracy*

The LDM estimates the position of the compromised sensor. One of the two inputs it takes is estimated sensor measurements from the FPDM. Hence the performance of the model in the LDM implies that one input is the predicted measurement using that model. We have adopted a multilabel classification technique where each sensor is classified as compromised or not. The macro-average precision, recall, and f1 value for the localization task is presented in Table 6. Interestingly all benchmark algorithms as well as the proposed XTM showed very close evaluation scores. CNN-Transformer model narrowly performs better than the XTM model in precision, recall, and f1 score. Our dataset is of multivariate nature with a feature dimension of size 54. Hence we introduce the row accuracy (RACC) metric. RACC measures the proportion of instances where every sensor location is correctly classified simultaneously, meaning for a time step, all compromised and uncompromised sensors are correctly classified. Table 7 shows that the Transformer model achieved the highest RACC, closely followed by XTM. The receiver operating characteristic curve (ROC) is made by checking the location detection accuracy for the different classification threshold values. The ROC and AUC curve for the LDM with XTM model estimation as input is shown in Figure 8. It can be observed that the XTM achieves a perfect area under the curve of 1.0, emphasizing the classification ability of the proposed technique.

**Table 7.** Performance comparison for detecting the location of FDI in IEEE-14 bus system.

| Model | Precision | Recall | f1-score | RACC |
|---|---|---|---|---|
| CNN | 0.9979 | 0.0.9981 | 0.0.9980 | 0.9227 |
| CNN-Transformer | 0.9989 | 0.9984 | 0.9986 | 0.9312 |
| Transformer | 0.9982 | 0.9981 | 0.9981 | 0.9396 |
| CNN-LSTM | 0.9983 | 0.9982 | 0.9982 | 0.9167 |
| Transformer-LSTM | 0.9984 | 0.9985 | 0.9984 | 0.9299 |

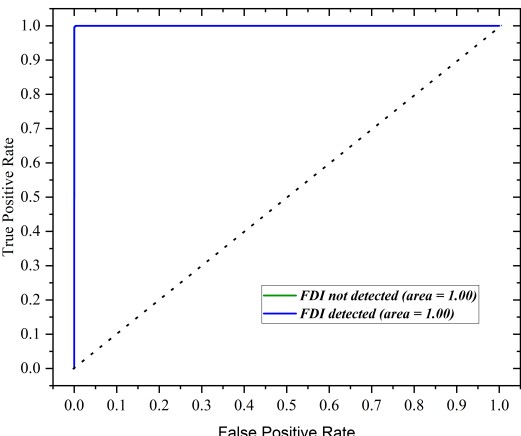

**Figure 8.** ROC AUC curve of location detection module with XTM.

### 6.5. Performance Evaluation on Minute Dataset

We have also trained and evaluated the performance of the proposed algorithm on the prepared minutely dataset. Table 8 shows the percentile values for choosing the threshold to detect false data injection presence. The FPDM trained on the minute data appeared to present the lowest threshold than any model trained on hourly data. It allows even more stealthier attacks to detect. This performance increase is natural since the deep learning models perform better on a larger dataset, which is especially true for transformer-based architectures. Training models with high sample rate and large training set increases the overall accuracy. Table 9 verifies this statement as it can be observed that using the minute dataset reduced the loss scores. The proposed algorithm has achieved RMSE, MSE and MAE score of 0.5912, 0.5956 and 0.4888 respectively. The receiver operating characteristic curve with its area is shown in Figure 9. The location detection model has shown highly precise classification ability achieving an area of 1.0 under the ROC curve.

**Table 8.** f1-score at different threshold value for proposed model trained with minutely data.

| Percentile (%) | Error Magnitude | f1-Score |
|---|---|---|
| 95 | 0.14308 | 0.97501 |
| 96 | 0.14545 | 0.97999 |
| 97 | 0.14914 | 0.98498 |
| 98 | 0.15309 | 0.99001 |
| 99 | 0.15893 | 0.99499 |
| 100 | 0.18707 | 0.99998 |
| Proposed $\tau$ | 0.2 | 1.00000 |

**Table 9.** Performance of Proposed Algorithm trained on Minutely Data.

| Metrics | Scores |
|---|---|
| RMSE | 0.5912 |
| MSE | 0.5956 |
| MAE | 0.4888 |
| f1 score of presence detection module | 1.0 |
| f1 score of location detection module | 1.0 |
| Row Accuracy | 0.9999 |

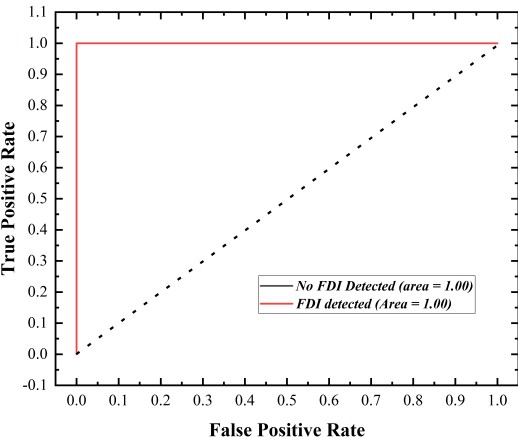

**Figure 9.** The ROC-AUC curve for proposed algorithm trained on minutely dataset.

## 7. Case Study

In this section, we have conducted an investigation on two particular attack cases. The case study illustrates the specifics, for instance buses that need to be targeted, the number of states as well as the states that need to be altered to launch FDI attacks. Finally, how our proposed model performs in those attack scenarios is analyzed. We have considered the standard IEEE 14 bus system for our case study, as shown in Figure 10. The system has the following parameters.

- Total number of lines = 20
- Total number of buses = 14
- Maximum number of measurements/sensors = 54 (i.e., 2 * 20 + 14)

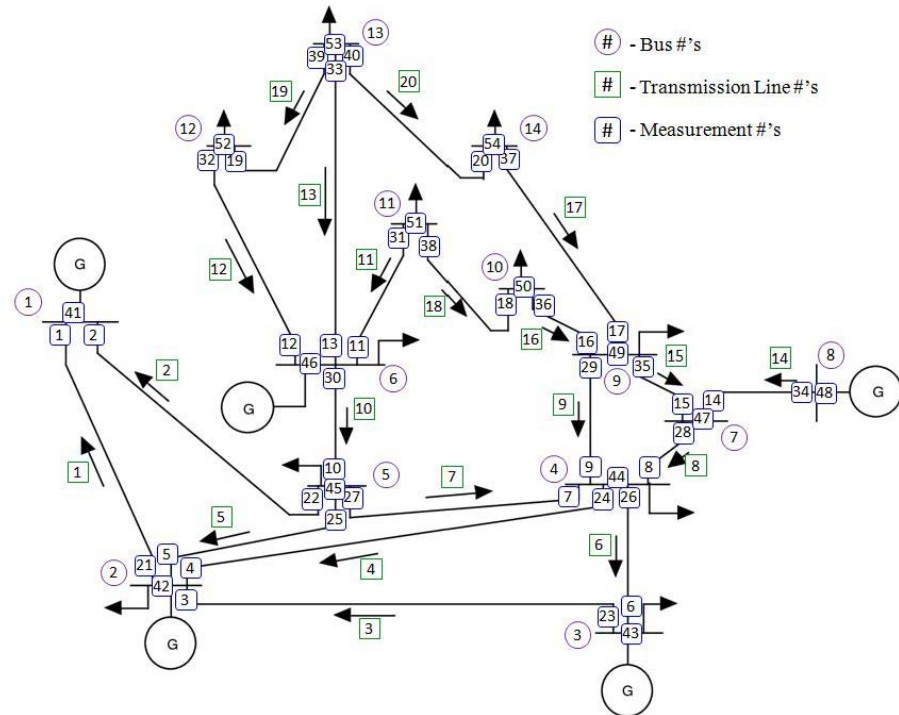

**Figure 10.** Standard IEEE 14 bus system [18].

Here, the first 20 measurements (1–20) are regarded as the forward line power flow, the reverse power flow of these lines is represented by the following 20 measurements (21–40) and the last 14 measurements (41–54) indicate the 14 buses' power consumption. We assume

that the attacker has access to manipulate 13 buses among 14 and at a time maximum of 5 buses along with 25 measurements could be altered. Target buses with corresponding states, along with the performance of the proposed algorithm for the particular attack cases are listed in Table 10.

**Table 10.** Details about Case Scenarios.

| Case Study Number | Number of Attacked Buses | Targeted Buses | Measurements Need to Be Changed | The Buses at Which One or More Measurements Are Required to Alter | FDI Detection Accuracy | Location Detection Accuracy |
|---|---|---|---|---|---|---|
| 1 | 2 | 12, 13 | 12, 13, 19, 20, 32, 33, 39, 40, 46, 52, 53, 54 | 6, 12, 13, 14 | 1.0 | 1.0 |
| 2 | 3 | 12,13,14 | 12, 13, 17, 19, 20, 32, 33, 37, 39, 40, 46, 49, 52, 53, 54 | 6, 9, 12, 13, 14 | 1.0 | 1.0 |

### 7.1. Case Study 1

Since it is considered that none of the measurements are secure in this situation, the attacker has access to all 54 measurements and can therefore introduce fake data into any of them. In this case, we have considered that the attacker can attack a maximum of two buses that means the measurements which need to be attacked are distributed in maximum of two buses. As a practical matter, attacking the state of any bus requires changing the power flow of the lines that are interconnected to that bus. The measurements of the buses must also be altered because those lines are attached to at least two separate nearby buses. In this example, the attacker's goal is to launch a successful FDI attack on 12 and 13 number buses. After executing a successful FDI attack, we have shown how efficiently XTM can detect this FDI attack. Moreover, it is also depicted that our proposed algorithm can accurately find out the exact attack location. For this case study, the following steps have been taken:

- It is necessary to conduct a FDI attack on buses 12 and 13 in order to accomplish the attacker's objectives.
- Measurements 12, 13, 19, 20, 32, 33, 39, 40, 46, 52, 53 and 54 must be altered to prevent the attack from being detected by BDD. Since they are all distributed among buses 6, 12, 13 and 14 the attacker must inject false data into those bus sensors.
- The attack vectors used in this study has the dimension of ($128 \times 54$). Here, 54 is the feature size of attack vectors that represents the changes to be made in each 54 sensors. Only the measurements that needed to be changed have non-zero values in any attack vector. We use one particular attack vector that specifically alter the measurements mentioned earlier to target bus 12 and 13.
- The selected attack vector is injected with the benign hourly test set. To detect FDI, first the compromised dataset is passed through FPDM. The proposed XTM is able to detect FDI in all measurements effectively using the threshold defined in Section 5.2.
- To detect the exact intrusion point, in other words the measurements that are altered, we utilize LDM. The forecasted measurements are concatenated with the injected sensor values and passed through a multilabel classifier. The proposed architecture is able to achieve 100% accuracy, detecting all the compromised sensors simultaneously.

### 7.2. Case Study 2

The attributes used in the case study 1 are maintained same in this example, with the exception of an improvement in the attacker's ability to attack the measurements. In this case the attacker has the ability to attack three buses targeting buses 12, 13 and 14. The following steps have been taken in this case:

- It is necessary to conduct a FDI attack on buses 12, 13 and 14 in order to fulfill the attacker's goal.
- Measurements 12, 13, 17, 19, 20, 32, 33, 37, 39, 40, 46, 49, 52, 53 and 54 need to be changed to avoid the attack from being noticed. Since they are all distributed

among buses 6, 9, 12, 13 and 14, the attacker should add malicious data into those measurements as well. The measurements in the benign hourly test data are changed accordingly to simulate this particular FDI attack.

- Similar to case study 1, FPDM is used to detect FDI for this particular attack case. XTM is able to detect intrusion with 100% accuracy even when the target buses increased.
- In the same way, the attacked measurements are detected through LDM utilizing the forecast from FPDM and current sensor readings. Similar to the previous example, the model is able to detect all measurements targeting bus 12, 13 and 14 effectively.

## 8. Conclusions

This paper presents a novel framework named XTM for detecting and locating FDI attacks in smart grids. FDI intrusion in the modern power system can bypass conventional BDD, causing catastrophic power failures while damaging infrastructures. Thus, research addressing the need for a robust and efficient FDI detector is of paramount importance. In this study, we have introduced a data-driven solution utilizing the Transformer algorithm from NLP that is trained on both one-year hourly and minutely sensor measurements in the IEEE-14 bus system. The proposed algorithm in this study can sense and locate FDI in real time utilizing two modules. To begin with, FPDM forecasts sensor measurements by looking at past data to find the deviation from the actual sensor readings. After that, the vector magnitude of this loss is compared with a threshold to detect FDI in the system. A new threshold selection scheme is introduced in this module that replaces the need for BDD, achieving a superior accuracy compared to the most commonly used models in the literature, for instance, CNN, LSTM and CNN-LSTM, when trained on hourly data. The forecasting ability of XTM can be inferred from the low RMSE score of only 1.26% when trained on the hourly dataset. The real-time forecast can also be utilized in FDI attack mitigation. In the next step, the intrusion location is detected by LDM in a multi-label fashion while taking both the actual and predicted sensor readings from the previous module as inputs. The proposed method was subjected to extensive training and evaluation that resulted in the average f1-score of 99.84% in all features with a very satisfactory row accuracy of 93.96%. In this case, row accuracy refers to the detection ability of the model in classifying all sensors at the same time. Additionally, when the model is trained on the minutely dataset, all the performance matrices experience dramatic increases, with row accuracy reaching almost 100%. This verifies our model's scalability and better prediction ability, if the training dataset is bigger. An elaborate discussion and case investigation regarding the formally built attack vectors demonstrate how attacking any bus requires changing multiple sensor measurements in the topology. Despite the superior performance of the proposed method, if any attacker increases the magnitude of FDI very slowly, then it might not be able to detect the attack. Thus in the future, we are interested in investigating how these slow-moving attacks can be detected using XTM. The dataset and the attack vectors are made publicly available to motivate further research.

**Author Contributions:** Conceptualization, A.B. ; methodology, A.B. and G.C.S.; software, A.B., G.C.S. and P.K.S.; validation, A.B., P.K.S., V.P.Y. and A.A.; formal analysis, A.B.; investigation, A.B. and A.A.; resources, G.C.S., V.P.Y. and A.A.; data curation, A.B. and A.A.; writing—original draft preparation, A.B. and G.C.S.; writing—review and editing, A.B., P.K.S., V.P.Y. and A.A.; visualization, A.A. and V.P.Y.; supervision, V.P.Y. and A.A.; project administration, A.B.; funding acquisition, A.A. All authors have read and agreed to the published version of the manuscript.

**Funding:** This research received no external funding.

**Institutional Review Board Statement:** Not applicable.

**Informed Consent Statement:** Not applicable.

**Data Availability Statement:** The hourly historical dataset is collected from the paper in [17], whereas the attack vectors are generated following the work in [18]. All dataset and the codes used in this study are openly accessible from the github repository (https://github.com/gcsarker/XTM).

**Conflicts of Interest:** The authors declare no conflict of interest.

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
