# Peer review of "XTM: A Novel Transformer and LSTM-Based Model for Detection and Localization of Formally Verified FDI Attack in Smart Grid"

_electronics, doi:10.3390/electronics12040797_

Round 1
Reviewer 1 Report
Interesting and timely required research study.
English writing requires extensive enhancements.
The related work section needs to contain more studies, the main shortage in each study that motivate this work to be introduced is required in this section. How different the proposed work from previous studies should be clearly addressed here. A table that summarizes previous studies' main contributions, used models, etc. is required in this section.
The two used datasets in the system should be clearly discussed, how and why they have been chosen.
The case study needs to be discussed more clearly, how it is different than the evaluation section.
Reviewer 2 Report
“XTM : A Novel Transformer & LSTM based model to Detect & Localization of Formally Verified FDI Attack In Smart Grid”
Date:12/29/2022
Document: electronics-2144531 including the cover page 22 pages.
The paper tackles an important topic about identifying and detecting FDI attacks in CPS. However, several issues need to be addressed in the manuscript. some of the comments are listed below:
- Please add some descriptions about “Transformer”. You are talking about power systems and the transformer term might mislead the reader.
- Please revise thoroughly figure 1, with the following suggestions:
- The state estimator is a part of the EMS.
- This system represents a centralized control system where one control center is controlling the whole system. I haven't seen such a system. The system is usually owned by different entities. You can add that to the caption and discuss it in the text. I can see several papers discussing the smart grid from Electronics MDPI, see this work (DOI:10.3390/electronics10091043)
- Many symbols from the equations need to be defined. Don’t expect the reviewer to know these symbols.
- You only need to define the acronyms once in the body of the manuscript e.g. (SCADA), (LSTM)….etc.
- It is preferred to summarize the related works in a taxonomy table which helps the reader to have a comparison between the other work findings.
- Please confirm if “H” is the measurement matrix or the Jacobian. You have mentioned both differentiations for the same symbol (see 211)
- It is almost impossible for the attacker to have full knowledge of H. A stealthy attack can successfully hack the system if it can manipulate some of the measurements, especially the critical ones. Please revise and support with references (see 211)
- It is recommended to add more case studies and attacks, especially if there is an attack on the system topology which is usually considered in such studies. See the following work (DOI:10.3390/en14237847).
- In equation (20), I assumed you represent the power balance set. Have you accounted for the converter power losses? If not, please explain.
- The whole manuscript needs to be grammatically revised; I can spot plenty of mistakes. For e.g., the title has extra space (XMT : ) and (In) in Smart Grid.
- You need to support several arguments with up-to-date references.
Round 2
Reviewer 1 Report
I think the paper can be accepted now
Author Response
Thank you again for your valuable time to provide recommendations. We have incorporated the feedback as per our best understanding.
Reviewer 2 Report
Thank you for your response, the manuscript looks good. I have two minor things.
Reference [36] is not fully cited or needs to have a second look
I am wondering about some of the old citations, I suggest having more recent publications as this field need an up to date literature. I have suggested a couple of newer publications, you can check them for your work.
